# Potent Anti-Cancer Activity of 1-Dehydrodiosgenone from the Product of Microbial Transformation of Steroid Saponins

**DOI:** 10.3390/ijms252313118

**Published:** 2024-12-06

**Authors:** Quanshun Li, Shirong Feng, Yuanyuan Zhang, Fangyuan Mou, Ting Guo, Baofu Qin, Yihan Liu

**Affiliations:** 1College of Life Sciences, Northwest A&F University, Xianyang 712100, China; liquanshun@nwafu.edu.cn (Q.L.); fengshirong@nwafu.edu.cn (S.F.); zhangyuanyuan405@163.com (Y.Z.); fangyuanmou@nwafu.edu.cn (F.M.); guoting@nwafu.edu.cn (T.G.); 2College of Chemistry and Pharmacy, Northwest A&F University, Xianyang 712100, China

**Keywords:** solid-state fermentation, microbial transformation, anti-cancer activity, apoptosis, safety test

## Abstract

Steroids are extensively used in the pharmaceutical industry as industrial raw materials for the production of anti-inflammatory and anti-tumor drugs. Microbial transformation, an environmentally friendly method, displays the potential for preparing steroids on an industrial scale. In this study, four steroids, including Diosgenin, Smilagenone, Yamogenin, and 1-Dehydrodiosgenone, were isolated and identified from the solid-state fermentation (SSF) product of a novel *Fusarium oxysporum* strain, and their anti-tumor activities were investigated. The cytotoxicity assay showed that 1-Dehydrodiosgenone had significant inhibitory effects on three tumor cell lines, Hala, A549, and Mad-MB468 cells, with IC_50_s of 6.59 μM, 5.43 μM, and 4.81 μM, respectively. 1-Dehydrodiosgenone significantly induced apoptosis and necrosis of Hala, A549, and Mad-MB468 cells by upregulating the expressions of cleaved caspase-3, cleaved PARP, Bax, and Bad. Moreover, no significant organ damage was observed in mice based on safety tests. Therefore, 1-Dehydrodiosgenone is expected to be developed as a safe and broad-spectrum anti-cancer agent.

## 1. Introduction

Steroids are important precursors for the synthesis of hormones and corticosteroids [1,2]. Steroids play crucial roles in physiological and pathological processes, such as anti-tumor [3,4], anti-inflammation [5], glucose regulation [6,7], and immune modulation [8]. With an estimated 9.7 million deaths each year, cancer remains a major global health problem [9,10]. Anti-cancer drugs tend to produce serious side effects due to high cytotoxicity and low selectivity [11]. These challenges drive researchers to search for novel small molecule compounds with potent anti-tumor activity and minimal side effects. Steroids are highly lipophilic and readily enter most cells to interact with intracellular receptors, making them ideal vehicles for targeting a broad array of pathologies [12]. Therefore, the development of steroids and derivatives has been a hotspot in the discovery of new cancer drugs [13,14].

Currently, steroids are mainly obtained from plants, and the major problem with this source is the extremely low yields and high consumption [15]. Traditionally, the production of steroids relied heavily on acid hydrolysis, typically using sulfuric and hydrochloric acids as catalysts. This process not only generates substantial amounts of acidic waste but also leads to serious environmental pollution. In recent years, with increasingly stringent global environmental regulations, many traditional steroid-processing companies have been forced to shut down due to non-compliance with environmental standards [16,17,18]. To address this problem, researchers have proposed a variety of improvement methods, including acid recycling [19], ionic liquids to replace the use of conventional acids [20], solid acid process catalysis [21], and photocatalysis [22]. Although these improvements have reduced pollution to some extent, there are still some technical bottlenecks. For example, the preparation of ionic liquids and solid acids is complicated, and the thermal hydrolysis reaction must be carried out at high temperature and pressure, resulting in high energy consumption. Photocatalysis suffers from slower reaction rates, lower catalyst efficiency, and greater dependence on light sources. Moreover, issues such as high costs and low efficiency have restricted the large-scale industrial application of these methods. Against this backdrop, microbial fermentation technology, with its mild reaction conditions, low energy requirements, and environmental friendliness, has gradually become an attractive alternative for producing steroids [23,24,25]. Compared to traditional acid hydrolysis, microbial fermentation effectively prevents the generation of significant volumes of acidic wastewater and can increase the yield of steroids by adjusting fermentation conditions and utilizing endogenous metabolic enzymes. It has been shown that bioconversion of cortisone using *Rhodococcus rhodnii* DSM 43960 can produce two new steroids, both with potential biological and therapeutic activities [26]. With the development of genetic and metabolic engineering, more studies are directed towards the production of steroids by microbial transformation. Through genetic modification and metabolic optimization, microbial strains are able to efficiently convert these phytosterols into intermediates that can ultimately be used for steroid production. [27]. Studies indicate that *Fusarium spinosum* spp. in solid-state fermentation (SSF) can effectively enhance steroid yield and abundance [28]. The excellent production potential of the *Fusarium spinosum* spp. has attracted widespread attention from researchers.

In this study, we successfully isolated an endophytic *Fusarium oxysporum* strain from the *Dioscorea rhizome* and named it SY_fxl_23.3. Systemic steroidal precursors with broad-spectrum anti-tumor activity and low toxicity were produced using SY_fxl_23.3 and SSF. We screened and identified four steroidal compounds and analyzed their chemical properties. The anti-tumor activity was screened using cellular assays and focused on the potential application of 1-Dehydrodiosgenone. In addition, the safety of the compound was assessed to ensure that its therapeutic potential would not result in toxic side effects. This study provides a theoretical basis for the application of 1-Dehydrodiosgenone as an anti-tumor drug precursor and to promote it as a potential anti-tumor drug candidate.

## 2. Results

### 2.1. Isolation and Identification of Strains

After 36 h of culture, the center of the colony was thicker, and the edges were thinner, with an overall white appearance. The surface of the colony was velvety, forming a cluster with a diameter of 2 to 3 cm with neat edges (Figure 1a). After 72 h, the hyphae grew outward and covered the entire Petri dish, with the hyphae mainly being white in color (Figure 1b). Spores of the strain were stained with lactophenol cotton blue and observed under an optical microscope (400×), showing a typical fusiform shape with tapered ends and a swollen middle (Figure 1c). Under scanning electron microscopy at 1.5k magnification, the hyphae appeared as a series of slender, linear structures with a surface featuring numerous folds (Figure 1d) At 13k and 25k magnification, the cell formation process was observed (Figure 1e,f). New cells primarily formed through budding from the lateral wall of the hyphae, which were relatively smooth, mostly rod- or fusiform-shaped.

Based on ITS region gene sequence analysis and BLAST database (NCBI) search, the strain was identified as *Fusarium oxysporum* (GenBank: OR707928) with a homology of 99.83%. Comparison with previously reported sequences in the NCBI database suggested that this may be a novel strain, named ‘SY_fxl_23.3’. The sample specimen was preserved at the China General Microbiological Culture Collection Center (CGMCC), No. 40905.

The quality of seed liquid prior to SSF directly affects the efficiency of subsequent biotransformation [29]. Mycelium fungal seed liquid with uniform particle size is the best choice for the next step of SSF (Appendix A). In the early stage of SSF, the SY_fxl_23.3 strain began to proliferate rapidly in the solid substrate, and many white colonies were distributed on the surface of the substrate (Appendix A). In the later stage of SSF (Appendix A), the hyphae of SY_fxl_23.3 had markedly increased, covering most of the substrate area, with colonies merging to form large growth zones. The substrate color darkened, likely due to pigments produced by microbial metabolic activities [30,31]. The results indicate that strain SY_fxl_23.3 exhibited strong growth and colonization abilities in both flask and SSF. Its robust hyphal extension and coverage capabilities provided a distinct advantage in SSF.

### 2.2. Structural Elucidation of Compounds ***1**–**4***

Compound **1** was obtained as a white solid (Figure 2a). The sample mass of compound **1** was 1.3 g. The chromatographic status of the compounds was displayed on a silica gel plate (Figure 2b). Its molecular formula was confirmed as C_27_H_42_O_3_ based on HR-ESI–MS data at *m*/*z* 415.3209 [M + H]^+^ (Appendix A). The Fourier transform infrared (FTIR) spectrum displayed a hydroxyl group (3447 cm^−1^) and an olefinic bond (1601.8 cm^−1^) (Appendix A). The characteristic signals, an olefinic proton (*δ*_H_ 5.33, overlap) and four methyls [*δ*_H_ 0.78 (s), 0.78 (d, *J* = 6.4 Hz), 0.96 (d, *J* = 7.0 Hz), 1.02 (s)], were observed in the ^1^H-NMR spectrum (Appendix A). Furthermore, the chemical shifts (*δ*_H_ 1–2) were overlapped in the ^1^H-NMR spectrum. The ^13^C NMR and DEPT135° spectra contained 27 carbons, including 4 quaternary carbons (*δ*_C_ 140.9, 109.5, 40.4, 36.8), 9 methines (*δ*_C_ 121.6, 81.0, 71.9, 62.2, 56.6, 50.2, 41.7, 31.6, 30.4), 10 methylenes (*δ*_C_ 67.0, 42.4, 39.9, 37.3, 32.2, 32.0, 31.7, 31.5, 28.9, 21.0), and 4 methyls (*δ*_C_ 19.6, 17.3, 16.4, 14.7) (Appendix A). By the analysis of the characteristic signals, compound **1** should be a typical C_27_ steroid analog containing a double bond. The speculation was further supported by HSQC (Appendix A). Finally, compound **1** was determined to be Diosgenin by comparing NMR data and mass spectrometry information with the reported literature (Figure 2e) [32].

Compound **2** was isolated as a white solid (Figure 2a). The mass of the sample of compound **2** was 780 mg. Its molecular formula was C_27_H_42_O_3_ according to the *m*/*z* 415.32098 [M + H]^+^ in the HR-ESI–MS data (Appendix A). The Fourier transform infrared (FTIR) spectrum displayed a hydroxyl group (3349.3 cm^−1^) and an olefinic bond (1603.7 cm^−1^) (Appendix A). The characteristic signals, an olefinic proton (*δ*_H_ 5.33, s) and four methyls [*δ*_H_ 0.79 (s), 0.79 (m), 0.98 (d, *J* = 6.8 Hz), 0.96 (s)], were observed in the ^1^H-NMR spectrum (Appendix A). The ^13^C NMR and DEPT135° spectra contained 27 carbons, including 4 quaternary carbons (*δ*_C_ 141.2, 109.7, 40.2, 37.0), 9 methines (*δ*_C_ 121.8, 81.2, 72.1, 62.5, 56.9, 50.4, 42.0, 32.0, 30.7), 10 methylenes (*δ*_C_ 67.2, 42.7, 40.7, 37.6, 32.4, 32.2, 31.8, 31.8, 29.2, 21.3), and 4 methyls (*δ*_C_ 21.3, 17.5, 16.7, 14.9) (Appendix A, e). By the analysis of the characteristic signals, compound **2** should be a typical C_27_ steroid analog containing a double bond. The speculation was further supported by HSQC (Appendix A). Finally, compound **2** was determined to be Yamogenin by comparing NMR data and mass spectrometry information with the reported literature (Figure 2f) [33,34,35].

Compound **3**, a pale yellow crystal (Figure 2a), shared a molecular formula of C_27_H_42_O_3_ according to its HR-ESI–MS data, indicating 7 degrees of unsaturation (Appendix A). The mass of compound **3** was 3.8 g. The characteristic signals, four methyls [*δ*_H_ 0.79 (s), 0.97 (d, *J* = 6.9 Hz), 0.79 (d, *J* = 6.3 Hz), 1.03 (s)], were observed in the ^1^H-NMR spectrum (Appendix A). Additionally, the chemical shifts (*δ*_H_ 1−2) were overlapped in the ^1^H-NMR spectrum. The ^13^C NMR and DEPT135° spectra contained 27 carbons, including 4 quaternary carbons (*δ*_C_ 213.4, 109.4, 40.8, 35.1), 8 methines (*δ*_C_ 80.9, 62.3, 56.4, 44.3, 41.7, 40.9, 35.3, 30.4), 11 methylenes (*δ*_C_ 67.0, 42.5, 40.2, 37.3, 37.1, 31.9, 31.5, 28.9, 26.6, 26.2, 21.1), and 4 methyls (*δ*_C_ 22.8, 17.3, 16.6, 14.6) (Appendix A). The crystal structure of compound **3** was obtained by the X-single crystal diffraction method. The positions of all hydrogen atoms were determined by riding bridge mode and refined by isotropic refinement to obtain Appendix A. The single crystal of the compound belongs to the orthorhombic crystal system with the space group P212121; the crystal structure was refined using OLEX_2_ with R_1_ = 6.21, WR_2_ = 14.92%, and Flack parameter = 0.3 (5). The characteristic symmetry element in the crystal contains three 21-helix axes; one crystal cell contains four compound molecules. Each molecule contains 11 chiral carbon atoms, and the absolute configurations of the chiral carbons are, in order, C5 (R), C8 (R), C9 (S), C10 (R), C13 (S), C14 (S), C16 (S), C17 (R), C20 (S), C22 (R), and C25 (R) (Figure 2c). Consequently, by comparing NMR data, mass spectrometry information, infrared data, and single-crystal absolute configuration data in the SciFinder database, this compound **3** was confirmed to be Smilagenone (Figure 2g). The crystallographic data for Smilagenone were deposited at the Cambridge Crystallographic Data Center under the reference number CCDC-2340543.

Compound **4** was obtained as a white crystal (Figure 2a). Compound **4** had a mass of 760 mg. The HR-ESI–MS data of compound **4** showed an ion peak at *m*/*z* 411.2896 [M + H]^+^, which was used to establish its molecular formula as C_27_H_38_O_3_, indicating 7 degrees of unsaturation (Appendix A). The characteristic signals, three olefinic protons [(*δ*_H_ 7.03, d, *J* = 10.1 Hz), (*δ*_H_ 6.22, dd, *J* = 10.1,1.9 Hz), (*δ*_H_ 6.06, overlap)] and four methyls [*δ*_H_ 1.24 (s), 0.96 (d, *J* = 6.9 Hz), 0.84 (s), 0.78 (d, *J* = 6.4 Hz)], were observed in the ^1^H-NMR spectrum (Appendix A). Moreover, the chemical shifts (*δ*_H_ 1−2) were overlapped in the ^1^H-NMR spectrum. The ^13^C NMR and DEPT135° spectra contained 27 carbons, including 5 quaternary carbons (*δ*_C_ 186.4, 169.2, 109.4, 43.7, 40.8), 10 methines (*δ*_C_ 155.9, 127.6, 124.0, 80.6, 62.1, 55.3, 52.5, 41.7, 35.3, 30.4), 8 methylenes (*δ*_C_ 67.0, 39.6, 33.8, 32.9, 32.0, 31.4, 28.9, 22.8), and 4 methyls (*δ*_C_ 18.9, 17.2, 16.6, 14.6) (Appendix A). The crystal structure of compound **4** was obtained by the X-single crystal diffraction method. The collected single-crystal data were processed in the same way to obtain Appendix A. The single crystal of the compound molecule belongs to the monoclinic crystal system with the P212121 space group; the crystal structure was refined using OLEX_2_ with R_1_ = 3.13%, WR_2_ = 7.51%, and Flack parameter = 0.04 (19). The characteristic symmetry element in the crystal contains three 21-helix axes; a single crystal cell contains four compound molecules. A single compound molecule contains 10 chiral carbon atoms, and the absolute configurations of the chiral carbons are, in order, C8 (S), C9 (S), C10 (R), C13 (S), C14 (S), C16 (S), C17 (R), C20 (S), C22 (R), and C25 (R) (Figure 2d). Therefore, compound **4** was confirmed to be 1-Dehydrodiosgenone (Figure 2h) by comparing NMR data, mass spectrometry information, infrared data, and single-crystal data. Crystallographic data for 1-Dehydrodiosgenone were deposited at the Cambridge Crystallographic Data Center under reference number CCDC-2325701.

^13^C NMR (100 Hz) data and ^1^H NMR (400 Hz) data of the four compounds are summarized in Table 1 and Table 2.

### 2.3. Anti-Tumor Activity Assays of Compounds

#### 2.3.1. Compounds **1**–**4** Inhibit Proliferation of Three Cancer Cell Lines In Vitro

To investigate the anti-tumor activity of the four compounds, the cck-8 assay was used to detect the cytotoxicity of serial concentrations (0–50 μM) of compounds **1**–**4** on Hala, A549, and Mad-MB468 cells. The results showed that compounds **1**–**4** reduced the survival of the tested cancer cells in a dose-dependent manner (Figure 3a–c). We also examined the IC_50_ values for the cytotoxicity of compounds **1**–**4** against Hala, A549, and Mad-MB468 cells at various time points, and the results are shown in Figure 3d and Table 3. 1-Dehydrodiosgenone showed significant potency against Hala, A549, and Mad-MB468 cells with IC_50_ = 6.59 μM, 5.43 μM, and 4.81 μM, respectively. The related properties of 1-Dehydrodiosgenone were superior compared to Diosgenin. Yamogenin (IC_50_ = 22.23 μM) was comparable to Diosgenin, and Smilagenone (IC_50_ = 25.58 μM) had slightly lower cytotoxic activity than Diosgenin. In addition, we tested the toxicity of 1-Dehydrodiosgenone on human embryonic kidney cells (HEK9 cells). The results showed that 1-Dehydrodiosgenone did not show significant toxicity to HEK9 cells when administered at a concentration of 400 µM (Appendix A).

#### 2.3.2. Effect of 1-Dehydrodiosgenone on Tumor Cell Proliferation

To further assess whether 1-Dehydrodiosgenone induced apoptosis in Hala, A549, and Mad-MB468 cells, we used EdU staining to detect DNA synthesis in the cells to further assess the effect of the compounds on cell proliferation. The results showed that EdU red fluorescence was significantly reduced after 1-Dehydrodiosgenone treatment (Figure 4a–c). Figure 4d shows that the proliferation rates of the three tumor cell types were significantly inhibited, with fluorescence intensities decreasing by 16.1 AU, 19.5 AU, and 11.78 AU, respectively. 1-Dehydrodiosgenone had the most pronounced effect on the A549 cells, and there was no obvious change in DAPI staining results (blue fluorescence), while the merged images showed a dramatic decrease in the red fluorescence decrease. Research has shown that 1-Dehydrodiosgenone can effectively inhibit the proliferation of tumor cells. This finding not only strongly supports its potential as an anti-tumor drug precursor, but also provides a direction for further research on its biological mechanism.

#### 2.3.3. Inhibitory Effect of 1-Dehydrodiosgenone on Tumor Cells

Calcein AM/Propidium Iodide (AM/PI) double staining was used to observe the number and morphological changes of Hala, A549, and Mad-MB468 cells. The green fluorescence density of the three tumor cell lines was significantly reduced and the red fluorescence density was significantly increased after 1-Dehydrodiosgenone treatment (Figure 4e–g). Quantitative analysis results showed that the fluorescence intensities of the three types of tumor cells were enhanced by 18.23 AU, 21.43 AU, and 19.27 AU, respectively, and the number of apoptotic cells increased, while the number of viable cells decreased (Figure 4h). Similar to the results of Section 2.3.2, 1-Dehydrodiosgenone showed the best inhibitory effect on A549 cells. In addition, the red fluorescence indicated that the apoptotic cells shrunk in size, confirming that the cells were undergoing apoptosis and further verifying the results of the AM/PI staining.

To more accurately evaluate the apoptosis-inducing effect of 1-Dehydrodiosgenone on Hala, A549, and Mad-MB468 cells, the cells were treated with 10 μg/mL 1-Dehydrodiosgenone for 24 h. The cells were stained with Annexin V-FITC/PI, and apoptosis was detected by flow cytometry (Figure 4i). The apoptosis rate (including early and late apoptosis) of Hala cells increased from 1.865% in the control group to 29.16%, an increase of 27.295%. The percentage of apoptosis in A549 cells increased from 2.17% to 21.63%, an increase in apoptosis of 19.46%. The rate of cell apoptosis in Mad-MB468 cells increased by 15.958% (Figure 4j). The experimental structures all indicated that 1-Dehydrodiosgenone had significant pro-apoptotic effects in different tumor cell lines, suggesting a broad spectrum of anti-tumor activity.

#### 2.3.4. 1-Dehydrodiosgenone Upregulates Apoptosis-Related Proteins to Inhibit Cell Proliferation

We used Western blot to study the effects of 1-Dehydrodiosgenone on apoptosis-related factors (e.g., Bax, Bad) and apoptosis effectors (e.g., caspase-3, PARP) to further explore the molecular mechanisms by which 1-Dehydrodiosgenone inhibits cell proliferation and induces apoptosis. The results showed that 1-Dehydrodiosgenone markedly increased the protein expression levels of cleaved caspase-3, cleaved PARP, Bax, and Bad while decreasing the expression levels of caspase-3 and PARP (Figure 4k). Quantitative analysis using image J 2.1.0 software (Figure 4l–o) revealed considerable differences in the optical density of apoptosis-related factors and apoptosis effectors.

### 2.4. Histopathological Analysis of Mouse Organs

The organs of the control and experimental groups of mice were compared (Figure 5a), and the organ weights of the two groups were statistically analyzed using GraphPad Prism 9.5. The results showed that 1-Dehydrodiosgeninone had no obvious effect on organ weight (Figure 5b). In addition, H&E staining was used to observe histopathological changes (e.g., inflammation) in mice. Compared with the control group (Figure 5c), the myocardial fibers in the treatment group were neatly arranged, the structure was intact, and there were no obvious lesions. The hepatocytes were arranged normally, the sinusoidal structure was clear, and there was no inflammation or necrosis. The brain tissue had a clear structure and was normally arranged, with no evidence of inflammation or degeneration. The renal tubules and glomeruli were normal, with no obvious pathological changes. The spleen tissue had a clear structure, with obvious white and red edges in the medulla and no inflammation or necrosis. The intestinal mucosa was intact, with neatly arranged villi and no inflammation or ulcers. The gastric mucosa was also intact, with orderly arranged glands and no inflammation or ulcers. In short, intraperitoneal injection of 1-Dehydrodiosgeninone did not cause major pathological changes in the tissue structure of the mice’s major organs. These findings further revealed that 1-Dehydrodiosgeninone does not cause an inflammatory response or abnormal proliferation in normal cells, thus demonstrating its low cytotoxicity.

## 3. Discussion

In recent years, the method of producing various medicinal products through the SSF of fungi has attracted widespread attention. SSF using fungi such as *Aspergillus niger* can increase the content of total flavonoids and total phenols in the by-product of jackfruit and improve its antioxidant and hypoglycemic activities [36]. The SSF of plant materials using filamentous fungi has been widely used to produce high-value γ-linolenic acid and β-carotene in the pharmaceutical industry [37]. Lovastatin is an anti-cholesterol agent that was produced by *Aspergillus terreus* using SSF [38]. In this study, four steroids were produced by SSF of a novel endophytic *Fusarium oxysporum* strain SY_fxl_23.3, and we obtained 1.3 g of Diosgenin, 780 mg of Yamogenin, 3.8 g of Smilagenone, and 760 mg of 1-Dehydrodiosgenone by isolation and purification. The strain grew extremely well in a solid substrate and had a rich metabolic enzyme system, which could convert multiple compounds with medicinal value. Some researchers have now screened another *Fusarium* species and, by optimizing the medium and other additives, the new medium has higher Diosgenin productivity than the traditional medium [39]. The endophytic *Fusarium spinosum* fungus C39 was shown to be effective in converting saponins from Japanese yam into Diosgenin, with an increase in Diosgenin concentration of 62.67% during 15 days of solid-state fermentation, in addition to the identification of 32 compounds [40]. Similarly, the *Fusarium spinosum* strains we studied have great potential to increase production efficiency by further optimizing fermentation conditions and expanding the fermentation scale, providing strong support for steroidal bioactive resources. It also provides an important foundation and direction for future natural product production and drug development.

According to reports, Diosgenin and its derivatives have pharmacological values such as inducing apoptosis, reducing oxidative stress, and anti-neuroinflammation [41], and the activity of some derivatives is even better than that of Diosgenin, which is consistent with our research results. The SSF process of this strain SY_fxl_23.3 provides a material guarantee for the development of subsequent drugs. On the other hand, compared with chemical synthesis, SSF has the advantages of higher volumetric productivity, a simpler and more environmentally friendly process, lower energy consumption, and a steady increase in additional biologically active substances, which is closer to the natural microbial habitat [42].

Apoptosis of tumor cells is a hotspot in tumor research. Cancer is a disease in which cells multiply abnormally and lose the ability to repair themselves or to die. Many anti-tumor drugs exert their effects by inducing apoptosis or increasing the susceptibility of cancer cells to apoptosis [43]. This experiment focused on the ability of the apoptosis induction for the four sterols. The results showed that they exhibited strong cytotoxicity, which was time and dose dependent and had potential medicinal value. Among them, 1-Dehydrodiosgenone had the best anti-cancer effect, with an IC_50_ of 4.81 μM. Interestingly, 1-Dehydrodiosgenone has about 4-fold higher cytotoxic activity than Diosgenin (IC_50_ = 21.27 μM), which might be due to the low bioavailability, low solubility, and poor metabolic stability of Diosgenin in vivo, poor targeting, and drug tolerance issues affecting its anti-tumor effects [44]. Through EDU cell proliferation staining, AM/PI dual staining, and Annexin V-FITC/PI staining, we found that 1-Dehydrodiosgenone appreciably induces apoptosis of tumor cells in vitro. Observation of the morphology of apoptotic cells revealed that the cells were shrunken and compacted, and some showed signs of lysis. These characteristics indicate that apoptosis is occurring [45]. Apoptosis is marked by the involvement of various cysteine aspartate-specific proteases [46]; these enzymes hydrolyze specific aspartic acid residues in the substrate in their active form [47] and induce apoptosis [48]. Caspases are classified as initiators (caspase-8, -9, and -10) and effectors (caspase-3, -6, and -7) [47]. Once activated, cysteine-aspartic proteases acquire their active form and can activate executioner cysteine-aspartic proteases in a cascade, thereby inducing cell death. To investigate the molecular mechanism of 1-Dehydrodiosgenone-induced apoptosis, this study used Western blot to determine the expression of the apoptotic executor caspase-3. Thus, during the process of cell death after compound treatment, caspase-8 and -9 were activated to obtain active forms, which further activated caspase-3 and cleaved various intracellular substrates, resulting in apoptosis [49] (Figure 6, Pathway 1). In addition, cleaved caspase-3 is essential for certain processes associated with DNA cleavage [50]. The PARP pathway plays a key role in cellular DNA repair. When DNA is broken, the PARP protein recognizes the damaged site and initiates the repair process. It modifies itself and other proteins by catalyzing ADP-ribosylation, thereby attracting repair proteins to the damage site. During apoptosis, activated caspase-3 cleaves PARP, rendering it unable to repair and promote apoptosis [51]. In this sense, our results are consistent with those reported in the literature. Studies have shown that PARP expression is significantly reduced in Western blot studies, and 1-Dehydrodiosgenone can significantly inhibit the PARP pathway.

Currently, the main treatment for cancer is chemotherapy, which focuses on eliminating tumor cells by promoting the production of apoptotic proteins, thereby activating the intrinsic apoptotic pathway of apoptosis [52,53]. Bad and Bax are pro-apoptotic proteins of the Bcl-2 family that play a key role in the apoptosis process. When the cell receives an apoptotic signal, Bax is highly expressed and translocated from the cytoplasm to the outer mitochondrial membrane. Bax forms oligomers on the membrane, opening a channel that promotes the release of cytochrome c from the mitochondria into the cytoplasm, which further triggers the caspase cascade reaction [54]. Under non-apoptotic conditions, Bad is phosphorylated and binds to anti-apoptotic proteins (e.g., Bcl-2), thereby maintaining cell survival. Upon apoptotic signals, Bad loses phosphorylation, dissociates from anti-apoptotic proteins, and forms a channel in collaboration with Bax proteins to release cytochrome C [55]. Our results suggest that 1-Dehydrodiosgenone may act as an apoptotic signal factor, activating pro-apoptotic proteins such as Bad and Bax in the mitochondrial apoptotic pathway, initiating a caspase cascade amplification reaction, which in turn induces the cell to enter the apoptotic process (Figure 6, pathway 2). This finding leads to the conclusion that the apoptosis-inducing capacity of 1-Dehydrodiosgenone is the result of the interaction of multiple signal pathways, including the caspase activation pathway, the mitochondrial apoptosis pathway, and the PARP pathway. However, the anti-tumor activity of the drug targets both cancer cells and normal tissue cells, and this activity means that there are significant side effects. In this sense, we used a mouse safety test to detect the safety of 1-Dehydrodiosgenone on normal tissue cells. By observing the HE staining pathology of the tissues and organs of normal mice, it was concluded that 1-Dehydrodiosgenone has low or zero cytotoxic activity. In the treatment of cancer, antineoplastic drugs need to selectively exhibit low cytotoxic activity against non-cancer cells while inducing programmed apoptosis of cancer cells. It is worth knowing that 1-Dehydrodiosgenone is the compound with this property. The above studies have shown that 1-Dehydrodiosgenone plays an outstanding role in the treatment of cancer and is expected to become a promising candidate drug precursor.

## 4. Materials and Methods

### 4.1. Reagents and Strains Materials

An endophytic fungal strain, named SY_fxl_23.3 (deposit number: CGMCC 40905), was isolated from the rhizomes of the plant *Dioscorea* spp., which were purchased from Yongchun Agricultural Science and Technology Company (Ankang, Shaanxi, China). The surface of the *Dioscorea rhizomes* was first washed three times with sterile water. The *Dioscorea rhizomes* were immersed in 75% alcohol for 3 min. The *Dioscorea rhizome* was then removed on an ultra-clean bench and the *Dioscorea rhizome* was crushed with a mortar and pestle after the alcohol had evaporated. Finally, 10×, 10^2^×, 10^3^×, 10^4^×, 10^5^×, 10^6^×, 10^7^×, and 10^8^× were diluted with sterile water and inoculated (100 µL) onto potato *Dioscorea* xtrose agar (PDA) using the coated plate technique. Each gradient was repeated three times and incubated in a constant temperature incubator (28 °C) for 3–4 d. Purification was repeated until single colonies were obtained.

Genomic DNA was isolated using a DNA extraction kit (Solarbio Science & Technology Co., Ltd., Beijing, China) and used as a template for PCR amplification. Forward and reverse primers were synthesized by Beijing Qingke Biotechnology Co. (Beijing, China). The amplified products were subjected to Sanger bi-directional sequencing, and ITS1 primer (5 TCCGTAGGTGAACCTGCGG-3) and ITS4 primer (5-TCCTCCGCTTATTGATATATGC-3) were used to amplify the ITS gene, and PCR was used to amplify the ITS rDNA sequence. PCR products were sent to Beijing AuGCT Biotechnology Co., Ltd. (Beijing, China) for purification and sequencing. The obtained ITS sequences were compared against the NCBI Blast database. The morphology and structure of the colonies were analyzed using a field emission scanning electron microscope (No. 2021252903, Hitachi High-Tech Corporation, Tokyo, Japan) and an optical microscope (NIKON, Tokyo, Japan).

Primary antibodies, Bad, caspase-3, Bax, and PARP were purchased from Cell Signaling Technology (Danvers, MA, USA). The secondary antibodies, Goat Anti-Mouse IgG/HRP and Goat Anti-Rabbit IgG H&L (HRP) were purchased from Millipore (Billerica, MA, USA). Other experimental reagents were purchased from Beijing AoBo Biotechnology Co., Ltd. (Beijing, China).

All mice (57 BL/6 mice) were housed under standard laboratory conditions, with controlled temperature, humidity, and a 12 h light/dark cycle, and provided with ad libitum access to food and water. All operations were based on the Guide for the Care and Use of Laboratory Animals [56]. All experimental procedures were approved by the Northwest A&F University Animal Care and Use Committee (Approval code: IACUC2024-1120).

### 4.2. Solid-State Fermentation and Extraction

The preserved SY_fxl_23.3 strain was removed from the −80 °C freezer and activated on PDA medium in a fungal incubator at 32 °C for 36 h. The spore concentration was adjusted to 1 × 10^6^ cfu/mL with 0.1% Tween-80, and the spore suspension was stored at 4 °C. The seed medium (0.5% sucrose, 0.1% yeast extract, 0.8% *Dioscorea rhizome* powder, 0.075% KH_2_PO_4_, and 0.025% MgSO_4_·7H_2_O) was sterilized at 121 °C for 20 min, then inoculated with the spore suspension at a 5% (*v*/*v*) ratio and incubated at 33 °C and 140 rpm for 24 h. The seed culture was inoculated at a 20% (*v*/*v*) ratio into the SSF substrate (48.6% *Dioscorea rhizome* powder, 0.8% sucrose, 0.5% yeast extract, 0.1% KH_2_PO_4_, and 30% distilled water). The mixture was stirred thoroughly and transferred to a custom-designed shallow-stacked-tray solid-state fermenter (Figure 7). The fermentation was maintained at a constant temperature of 33 °C, and the product was obtained after 8 days of incubation.

The solid media (4.2 kg) were crushed and then extracted with petroleum ether. The extracts (53.0 g) were concentrated in vacuo. The extract was subjected to silica gel CC eluting with gradient petroleum ether/EtOAc (from 15:1 to 8:1) to afford five fractions (Fr. A–D). Further separation was performed using an ODS chromatography column, with monitoring and auxiliary separation achieved through a combination of vanillin staining and thin-layer chromatography (TLC). Fr. A (5.6 g) was further separated on an ODS column (petroleum ether–ethyl acetate, 12:1 to 5:1, *v*/*v*) and purified by RP-HPLC elution, yielding compound **1** (1.3 g, *t*_R_ 14 min, CH_3_CN–H_2_O, 80%). Fr. B (4.1 g) was loaded onto an ODS column (petroleum ether–ethyl acetate, 10:1 to 1:1, *v*/*v*) for further chromatographic separation, yielding four fractions (B1–B4). Fr. B2 (1.8 g) was then applied to an ODS column (petroleum ether–ethyl acetate, 10:1 to 5:1, *v*/*v*), resulting in compound **2** (780 mg, *t*_R_ 15.75 min, CH_3_CN–H_2_O, 80%). Fr. C (11.2 g) was loaded onto an ODS column (petroleum ether–ethyl acetate, 15:1 to 5:1, *v*/*v*) for further chromatographic separation, yielding compound **3** (3.8 g, *t*_R_ 9.7 min, CH₃CN_3_H_2_O, 85%). Fr. D (3.4 g) was loaded onto an ODS column (petroleum ether–ethyl acetate, 12:1 to 3:1, *v*/*v*) for further chromatographic separation, yielding five fractions (D1–D5). Fr. D3 (1.1 g) was then applied to an ODS column (petroleum ether–ethyl acetate, 10:1 to 5:1, *v*/*v*) and semipreparative HPLC (CH_3_CN/H_2_O, 75:25) to obtain compound **4** (760 mg, *t*_R_ = 10.2 min).

### 4.3. Identification of Compounds

^1^H NMR, ^13^C NMR, DEPT135°, and HSQC spectra were recorded on an NMR spectrometer (AVANCE NEO 400MHZ Switzerland, Bruker Switzerland Ltd., Fällanden, Switzerland). A Fourier transform infrared spectrometer (FTIR) Model: Vertex70; Ettlingen, Germany) was used to detect functional group information of the corresponding compounds. The molecular weights of the compounds were quantified in positive ion mode using a high-resolution ion mobility liquid-mass spectrometer (HRLCMS model: LC-30A + Triple TOF5600+; Singapore).

### 4.4. Single Crystal Analysis of Compounds ***3*** and ***4***

Methanol/water/chloroform = 8:2:1 was selected as the solvent to dissolve 500 mg of compounds **3** and **4**, which were fixed to 5 mL and passed through an organic filter membrane with a pore size of 0.45 microns. It was placed in a refrigerator at 4 °C and kept in a dry state to allow the solvent to evaporate slowly to obtain a supersaturated solution and to promote the growth of crystals. The solution was allowed to stand for two weeks, during which time no movement was allowed until crystals were precipitated. The crystal data were measured using a BRUKER D8 VENTURE diffractometer equipped with a cryogenic device using a Cu-targeted Kα (λ = 1.54178 Å) diffractometer with a test temperature of 100 K. The data were collected by φ-ω scanning. The collected single-crystal data were analyzed using the SHELXT program via direct methods to determine the compound structure. All data were integrated with SAINT, and multi-scan absorption corrections were applied using SADABS. The coordinates and thermal vibration parameters of all non-hydrogen atoms were refined anisotropically using full-matrix least squares methods in Olex2.

### 4.5. Anti-Tumor Activity Assay of Compounds

#### 4.5.1. Cell Culture

Hela, A549, and Mad-MB468 cell lines were purchased from the National Collection of Authenticated Cell Cultures (Shanghai, China). Three cell lines were cultured on high-sugar DMEM medium supplemented with 10% fetal bovine serum, 100 U/mL penicillin, and 100 μg/mL streptomycin in a humidified atmosphere with 5% CO_2_ at 37 °C.

#### 4.5.2. Cytotoxicity Assay

Stock solutions of compounds **1**–**4** were prepared in DMSO and concentration groups were set up at 0 μM, 2 μM, 4 μM, 8 μM, 16 μM, 32 μM, and 50 μM. The cells were seeded in a 96-well plate (3 × 10^4^ cells/well). After 24 h of incubation, the cells were treated with the compound concentrations mentioned above for 12 h, 24 h, 36 h, and 48 h, with each concentration tested in triplicate. When changing the medium, 10% CCK8-containing medium was added, and the cells were incubated at 37 °C for an additional 2 h. The optical density (OD) values at different time points were measured at 650 nm using the Bio-Tek ELX800 microplate reader, Winooski, VT, USA.

#### 4.5.3. Cell Proliferation Assay

The cell proliferation assay was completed using the BeyoClick™ EdU-555 cell proliferation detection kit (Beyotime, Shanghai, China). Three cell lines (Hela, A549, and Mad-MB468 cells) were incubated with different concentrations of 1-Dehydrodiosgenone for 24 h, with 0.1% DMSO serving as control. The EdU solution (100 μL) was added to the cell medium for 2 h. Cells were fixed with 4% paraformaldehyde for 15 min, incubated with 0.3% Triton X-100 permeabilization solution at room temperature for 10 min, and then dark incubated with Click reaction solution for 30 min at room temperature. Finally, 1× Hoechst 33342 solution was added, and the cells were incubated in the dark at room temperature for 10 min. After washing with PBS three times, fluorescence detection was performed using the MSHOT digital microscopic imaging system.

#### 4.5.4. Calcein AM/Propidium Iodide (AM/PI) Dual Staining

The Calcein/PI Cell Viability and Cytotoxicity Assay Kit (Beyotime, Shanghai, China) was used to measure cell viability. Calcein AM/PI (100 μL) was added to both the experimental and control groups, followed by incubation at 37 °C in the dark for 30 min. Staining results were observed under a fluorescence microscope (Axio imager. M2, Jena, Germany). Calcein AM emitted green fluorescence at Ex/Em = 494/517 nm, and PI emitted red fluorescence at Ex/Em = 535/617 nm.

#### 4.5.5. Apoptosis Detection

The ANNEXIN V-FITC/PI apoptosis detection kit (No. CA1020, Solarbio, Beijing, China) was used to analyze tumor cells via flow cytometry. In both the control and experimental group cells, 5 μL of Annexin V and PI staining solution was added, followed by incubation in the dark at room temperature for 15 min. The cells were centrifuged at 1000 rpm for 5 min at room temperature, the supernatant was removed, and the cells were resuspended in 500 μL of PBS. Samples detection was performed using a flow cytometer (BD FACSAria™ III, Franklin Lakes, NJ, USA).

#### 4.5.6. Western Blotting

After treatment of Hala, A549, and Mad-MB468 cell lines with 1-Dehydrodiosgenone for 24 h, the proteins were lysed in an ice bath in RIPA buffer for 30 min. The lysis mixture was centrifuged at 12,000 rpm for 10 min. The content of proteins contained in one part of the supernatant was detected with the BCA Protein Assay Kit according to the accompanying manual. The other part of the supernatant was diluted to be the same given concentration according to the above protein content.

The diluted protein sample of each group was mixed well with 5× loading buffer and heated in the metallic bath for 10 min, then cooled to room temperature. The cooled samples of all groups were separated by electrophoresis (10% SDS-PAGE) and then transferred to a PVDF membrane at 100 V for 2 h. The PVDF membrane was sealed with 5% skim milk at room temperature for 1 h and then washed three times with TBST for 10 min each time. The washed and cleaned membrane was incubated with primary antibodies (1:2000), including PARP and Bax, in TBST overnight at 4 °C. After washing three times, the membrane was incubated with secondary antibodies (1:2000) at room temperature for 1 h. Proteins were visualized using an ultra-sensitive ECL kit and quantified with image J 2.1.0 software. GAPDH was used as an internal reference.

### 4.6. Histopathological Analysis

All mice were divided into two groups of ten mice each. The mice of the first group were injected intraperitoneally with 100 μL of 1-Dehydrodiosgenone solution at a concentration of 10 μg/mL every 2 d, and the mice of the second group were injected with the same volume of DMSO solvent every 2 d as a control. The mice were put to death after 14 days, and the hearts, livers, brains, kidneys, spleens, intestines, and stomachs were taken out.

Each organ was weighed separately using a precision balance and the data were recorded. The weighing of each organ was repeated three times to ensure the accuracy of the data. To assess the effect of 1-Dehydrodiosgenone on the weights of major organs in mice, each tissue was collected for HE staining observation. After fixing the tissues, they were trimmed to 25 px × 25 px × 5 px. The tissues, fixed with 4% paraformaldehyde for 24 h, were dehydrated with alcohol, then embedded with the dissolved paraffin and solidified at −20 °C, dehydrated in alcohol, and dissolved in xylene for transparency. The tissues were sliced with wax blocks (4 μm thick for each slice), blanched, and pasted on slides, dried at 45 °C, baked at 60 °C for 30 min, and washed with xylene, gradient ethanol, and distilled water. The resulting sections were stained with hematoxylin and eosin (H & E), subsequently dehydrated with alcohol, and sealed with neutral gum. The histopathological changes (hearts, livers, brains, kidneys, spleens, intestines, stomachs, and degrees of lesions) of the stained colons were observed under the microscope (Mingmei, Hangzhou, China), and images were acquired and analyzed on the Mingmei Microscan Digital Imaging System.

### 4.7. Statistical Analysis

Statistical analyses were performed using GraphPad Prism 9 software, and all data were averaged from three independent experiments using one-way ANOVA or two-sided Student’s *t*-test. All data are expressed as mean ± SD, and *p* values less than 0.05 were considered statistically significant. ns, not statistically significance; * *p* < 0.05; ** *p* < 0.01; and *** *p* < 0.001.

## 5. Conclusions

In this research, Diosgenin, Smilagenone, Yamogenin, and 1-Dehydrodiosgenone were purified from the SSF products of *Fusarium oxysporum* SY_fxl_23.3. The broad-spectrum anti-tumor activity of 1-Dehydrodiosgenone was verified by cell proliferation experiments, cell death staining experiments, and cell apoptosis experiments. Western blot analysis confirmed that apoptosis and necrosis were related to the upregulation of apoptosis-related proteins Bax and Bad and the activation of the caspase cascade reaction. It was demonstrated that the anti-tumor activity of 1-Dehydrodiosgenone was mainly achieved through the endogenous apoptotic pathway (mitochondria), activation of the caspase pathway, and the PARP (poly-ADP-ribose polymerase) pathway. The results of the safety test in mice showed that 1-Dehydrodiosgenone had no obvious inflammatory reaction and toxic damage to the main organs of mice. 1-Dehydrodiosgenone has important potential as a precursor of broad-spectrum anti-tumor drugs. The development of an industrially feasible solid fermentation process to produce 1-Dehydrodiosgenone could provide raw material support for its application. In addition, due to the excellent production potential of this strain, its solid-state fermentation samples could be further screened for other biologically active steroids to further increase its economic value.

## Figures and Tables

**Figure 1 ijms-25-13118-f001:**
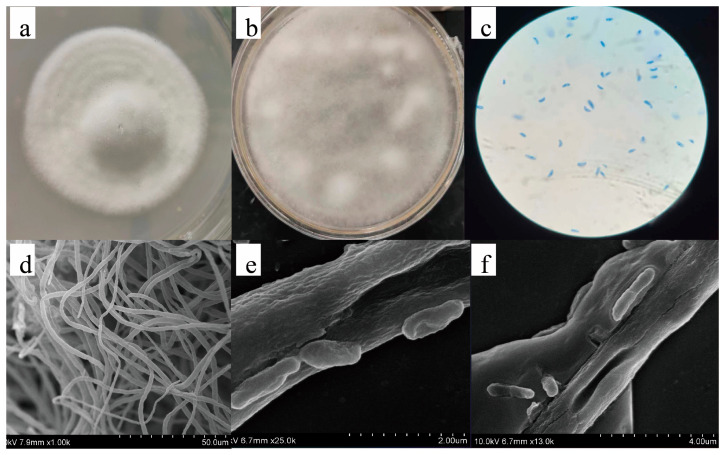
Morphological and microscopic identification of *Fusarium oxysporum* SY_fxl_23.3. (**a**) Colony morphology of SY_fxl_23.3 strain for 36 h of growth; (**b**) colony morphology of SY_fxl_23.3 strain for 72 h of growth; (**c**) spore lactophenol cotton orchid staining of SY_fxl_23.3 strain under light microscope (400×); (**d**) mycelium scanning electron microscope diagram of SY_fxl_23.3 strain (1k×); (**e**) mycelium scanning electron microscope diagram of SY_fxl_23.3 strain (25k×); (**f**) scanning electron micrograph of the proliferation mode of SY_fxl_23.3 strain (13k×).

**Figure 2 ijms-25-13118-f002:**
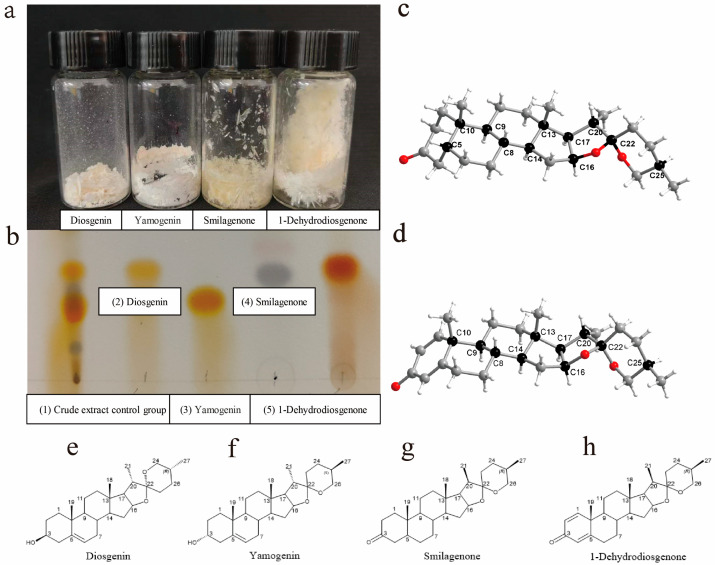
(**a**) Purified samples of four compounds; (**b**) thin-layer chromatogram of four compounds; (**c**) chiral carbon distribution of the Smilagenone molecule; (**d**) chiral carbon distribution of 1-Dehydrodiosgenone molecule; (**e**–**h**) schematic diagram of the molecular structure of four compounds.

**Figure 3 ijms-25-13118-f003:**
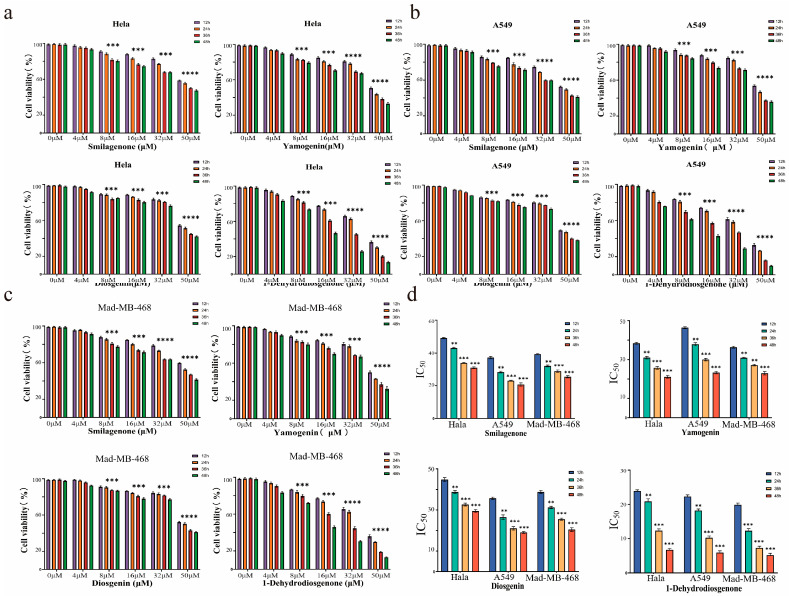
Antiproliferative effects of four compounds on Hala, A549, and Mad-MB468 cells. (**a**) Survival ratew (%) of Hala cells under different treatment concentrations of the four compounds with different action times; (**b**) survival rates (%) of A549 cells under different treatment concentrations of the four compounds with different action times; (**c**) survival rates (%) of Mad-MB468 cells under different treatment concentrations of the four compounds with different action times; (**d**) half inhibitory concentration (IC_50_) values of the four compounds on Hala, A549, and Mad-MB468 cells. Data are expressed as mean ± SD (** *p* < 0.01, *** *p* < 0.001, **** *p* < 0.0001).

**Figure 4 ijms-25-13118-f004:**
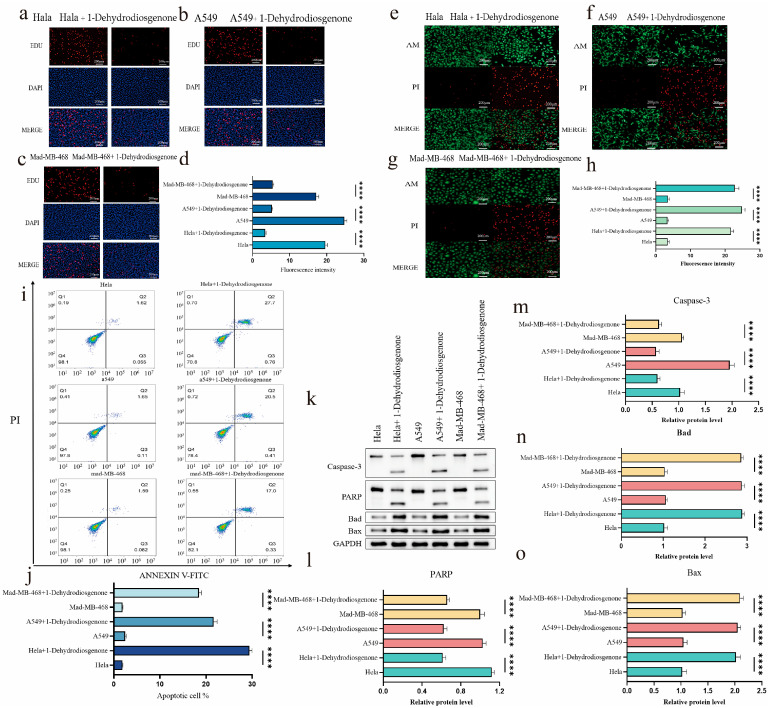
In vitro anti-tumor activity test of 1-Dehydrodiosgenone. (**a**–**c**) Comparison of proliferative fluorescence of EDU on Hala, A549, and Mad-MB468 cells before and after 1-Dehydrodiosgenone treatment; (**d**) quantitative analysis of fluorescence intensity of proliferating tumor cells using image J 2.1.0 software, represented as bar graphs; (**e**–**g**) fluorescence comparison of Hala, A549, and Mad-MB468 cells stained with AM/PI; (**h**) quantitative analysis of fluorescence intensity of AM/PI-stained apoptotic tumor cells using image J 2.1.0 software; (**i**) flow cytometry analysis of apoptosis percentage of Hala, A549, and Mad-MB468 cells before and after 1-Dehydrodiosgenone treatment; (**j**) the percentages of these cells are expressed as mean ± SD using columns; (**k**) Western blot detection of the expression levels of PARP, cleaved PARP, caspase-3, cleaved caspase-3, Bad, and Bad in tumor tissues; (**l**–**o**) quantitative optical density values from image J 2.1.0 software are presented as bar graphs. Data are expressed as mean ± SD (**** *p* < 0.0001).

**Figure 5 ijms-25-13118-f005:**
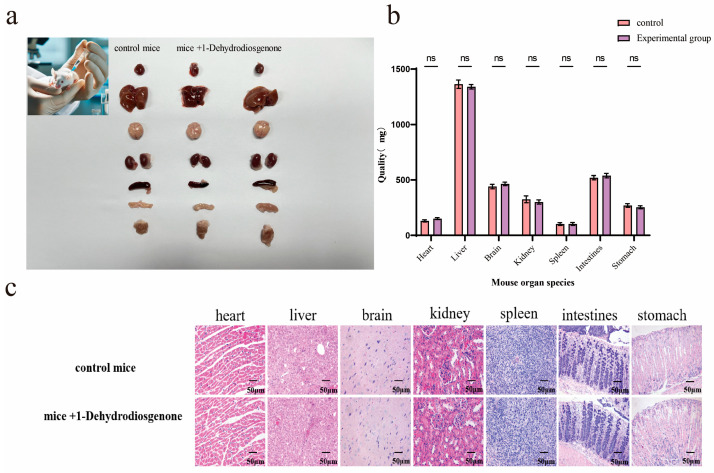
Safety test in mice. (**a**) Comparison of the dissected organs of the two groups of mice; (**b**) comparison of the weights of the dissected organs of the two groups of mice; (**c**) comparison of the HE staining of the tissue sections of the organs of the mice. The data are expressed as the mean ± SD (ns represents not significant).

**Figure 6 ijms-25-13118-f006:**
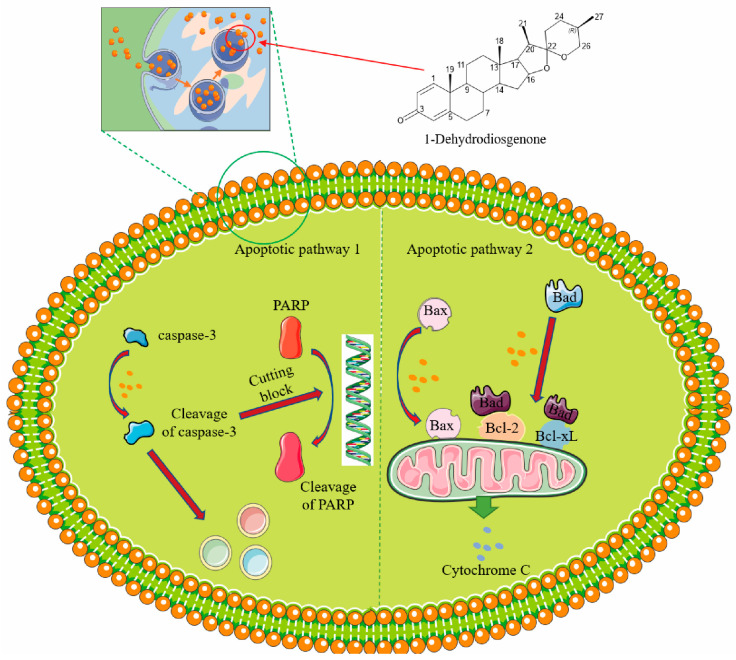
The anti-cancer pathway of 1-Dehydrodiosgenone.

**Figure 7 ijms-25-13118-f007:**
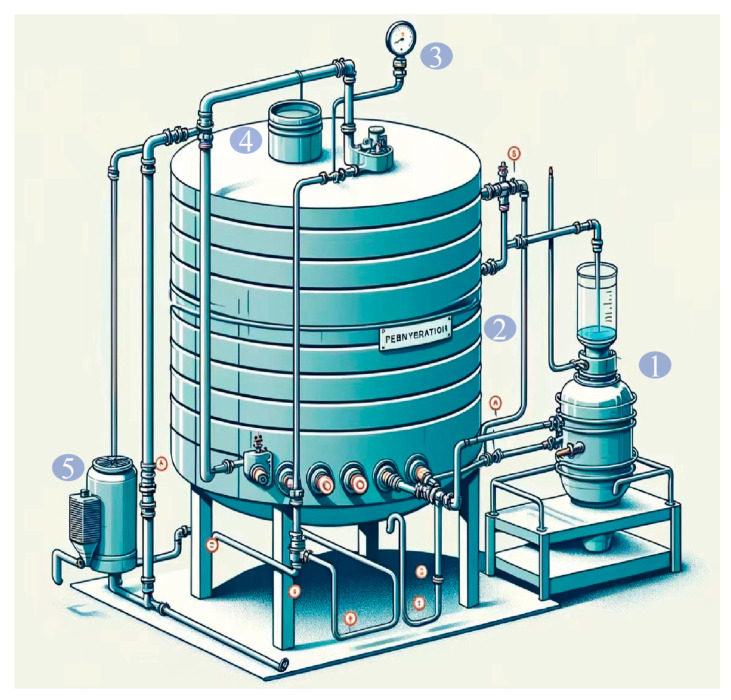
Schematic diagram of the SSF reactor. ①: humidification device; ②: substrate tray; ③: thermometer; ④: eyepiece; ⑤: cooling and temperature control device.

**Table 1 ijms-25-13118-t001:** The ^13^C NMR (100 Hz) data for compounds **1**–**4** (*δ* in ppm) in CDCl_3_.

No_._	Compound 1	Compound 2	Compound 3	Compound 4
1	37.3	37.6	37.3	155.9
2	31.5	32.4	37.1	124.0
3	71.9	72.1	213.4	186.4
4	42.4	42.7	42.5	127.6
5	140.9	141.2	44.3	169.2
6	121.6	121.8	26.2	33.8
7	32.2	32.2	26.6	32.0
8	31.6	32.0	35.3	35.3
9	50.2	50.4	40.9	52.5
10	36.8	37.0	35.1	43.7
11	21.0	21.3	21.1	22.8
12	39.9	40.7	40.2	39.6
13	40.4	40.2	40.8	40.8
14	56.6	56.9	56.4	55.3
15	32.0	31.8	31.9	32.9
16	81.0	81.2	80.9	80.6
17	62.2	62.5	62.3	62.1
18	16.4	16.7	16.6	16.6
19	19.6	21.3	22.8	18.9
20	41.7	42.0	41.7	41.7
21	14.7	14.9	14.6	14.6
22	109.5	109.7	109.4	109.4
23	31.7	31.8	31.5	31.4
24	67.0	29.2	28.9	28.9
25	30.4	30.7	30.4	30.4
26	28.9	67.2	67.0	67.0
27	17.3	17.5	17.3	17.2

**Table 2 ijms-25-13118-t002:** The ^1^H NMR (400 Hz) data for compounds **1**–**4** (*δ* in ppm, *J* in Hz) in CDCl_3_.

No.	Compound 1	Compound 2	Compound 3	Compound 4
1	*a* 1.83 m *b* 1.05 m	*a* 0.99 m *b* 1.82 m	*a* 2.32 dd, (14.7, 5.4) *b* 1.38 m	7.03 d (10.1)
2	1.62 (m)	*a* 1.43 m *b* 1.97 m	*a* 2.17 m *b* 2.02 m	6.06 overlap
3	3.52 m	3.44 m		
4	*a* 2.28 m *b* 2.22 m	*a* 2.27 m *b* 2.21 m	*a* 2.68 dd (15.1, 13.3) *b* 2.02 m	6.22 dd (10.1, 1.9)
5			1.78 m	
6	5.33 overlap	5.33 overlap	*a* 1.51 m *b* 1.09 m	*a* 1.93 m *b* 1.03 m
7	1.96 m	*a* 1.25 m *b* 1.43 m	*a* 1.89 m *b* 1.25 m	*a* 1.99 m *b* 1.31 m
8	1.64 m	1.56 m	1.65 m	1.79 m
9	0.93 m	0.87 m	1.46 m	1.06 m
10				
11	1.49 m	*a* 1.54 m *b* 1.61 m	1.45 m	1.66 m
12	*a* 1.72 m *b* 1.16 m	*a* 1.09 m *b* 1.24 m	*a* 1.74 m *b* 1.20 m	*a* 1.74 m *b* 1.15 m
13				
14	1.08 m	1.02 m	1.20 m	1.06 m
15	*a* 1.83 m *b* 1.27 m	*a* 1.64 m *b* 1.82 m	*a* 2.02 m *b* 1.25 m	*a* 2.48 tdd (13.5, 5.1, 1.6) *b* 2.35 ddd (13.3, 4.4, 2.5)
16	4.40 dd (14.4, 7.8)	4.39 dd (14.4, 7.8)	4.41 dd (14.3, 7.9)	4.39 dd (14.9, 7.6)
17	1.76 m	1.85 m	1.78 m	1.74 m
18	0.78 s	0.79 s	0.79 s	0.84 s
19	1.02 s	0.96 s	1.03 s	1.24 s
20	1.85 m	1.97 m	1.86 m	1.85 m
21	0.96 d (7.0)	0.98 d (7.0)	0.97 d (6.9)	0.96 d (6.9)
22				
23	*a* 1.61 m *b* 1.50 m	*a* 1.39 m *b* 1.59 m	1.62 m	*a* 1.99 m *b* 1.60 m
24	*a* 3.46 ddd (10.9, 4.6, 2.1) *b* 3.36 (t, 10.9)	*a* 1.37 m *b* 1.56 m	*a* 1.62 m *b* 1.45 m	*a* 1.60 m *b* 1.11 m
25	1.60 m	1.61 m	1.62 m	1.60 m
26	*a* 1.60 m *b* 1.45 m	*a* 3.37 m *b* 3.31 m	*a* 3.48 ddd (10.9, 4.5, 2.1) *b* 3.37 t (10.9)	*a* 3.46 dd (11.0, 2.9) *b* 3.35 t (11.0)
27	0.78 d (6.4)	0.79 d (6.4)	0.79 d (6.3)	0.78 d (6.4)

**Table 3 ijms-25-13118-t003:** Half inhibitory concentration (IC_50_) values of the four compounds on Hala, A549, and Mad-MB468 cells at different times.

		IC_50_ (μM)		
Time	Compounds	Hala	A549	Mad-MB-468
12 h	Diosgenin	43.83 ± 1.05	36.24 ± 1.73	38.98 ± 1.35
	Yamogenin	38.54 ± 0.96	46.58 ± 0.81	36.46 ± 1.17
	Smilagenone	48.93 ± 1.43	37.56 ± 0.54	39.76 ± 1.24
	1-Dehydrodiosgenone	24.34 ± 1.12	22.76 ± 0.98	20.42 ± 0.57
24 h	Diosgenin	38.24 ± 1.06	27.76 ± 1.22	31.67 ± 0.53
	Yamogenin	30.25 ± 2.13	38.76 ± 1.37	30.79 ± 0.71
	Smilagenone	42.87 ± 0.73	28.34 ± 0.76	31.76 ± 0.83
	1-Dehydrodiosgenone	20.34 ± 1.45	18.49 ± 1.42	12.48 ± 1.36
36 h	Diosgenin	31.89 ± 1.37	20.19 ± 0.55	25.63 ± 1.75
	Yamogenin	25.46 ± 2.23	30.76 ± 0.76	27.56 ± 0.85
	Smilagenone	33.76 ± 0.92	23.19 ± 1.19	28.46 ± 0.97
	1-Dehydrodiosgenone	12.46 ± 0.74	10.76 ± 1.76	7.83 ± 0.83
48 h	Diosgenin	28.76 ± 0.57	18.75 ± 2.13	20.67 ± 0.73
	Yamogenin	21.43 ± 1.07	23.57 ± 1.20	23.88 ± 0.49
	Smilagenone	30.43 ± 2.43	20.76 ± 0.59	25.76 ± 1.21
	1-Dehydrodiosgenone	6.59 ± 1.75	5.43 ± 0.77	4.81 ± 1.26

## Data Availability

All experimental data are included in this paper and the Appendix A.

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
