# Peer review of "Potent Anti-Cancer Activity of 1-Dehydrodiosgenone from the Product of Microbial Transformation of Steroid Saponins"

_ijms, 2024, doi:10.3390/ijms252313118_

Round 1
Reviewer 1 Report
Comments and Suggestions for Authors
The study employs a comprehensive approach, including advanced analytical techniques such as NMR, FTIR, and single-crystal diffraction, along with cytotoxicity assays, apoptosis induction analysis, and in vivo safety tests. While the article represents a significant way of stepping forward in developing environmentally friendly methods for producing anticancer compounds. However, the article needs a very minor corrections or revision.
Line 55-56
“Studies indicate that this strain in SSF can effectively enhance steroid yield and abundance.” Which study? Please provide references.
Line 56-57
“The excellent production potential of the strain SYfxl23.3 has attracted widespread attention from researchers.”
According to the author this strain might be a new one [Line 78-80]; how this specific strain has attracted other researchers? Please clarify.
Line 192-194
“The concentration of the drug that inhibits 50% (IC50) values of compounds 1–4 on Hala, A549, and Mad-mb468 cells at different times of action was calculated and shown in Figure 3d and Table 3.”- Please rephrase it as it’s hard to understand.
Line 305
The strain name is wrong; please correct it
Line 307-309
“In addition, it also contains a variety of economically beneficial products such as β-sitosterol, campesterol, and stigmasterol.”- Please justify your statement.
Optimizing fermentation condition
The focus on green chemistry principles, such as the use of the Fusarium oxysporum strain SYfxl23.3 in solid-state fermentation, marks a progressive approach in pharmaceutical manufacturing. However, while the strain's potential is demonstrated, there is limited discussion on optimizing and scaling this fermentation process. Please make a remark.
Safety profile
For safety profile, during the cytotoxic and antiproliferative assay experiment, I wonder whether the author tested isolated compounds against any normal healthy cell lines or not. If they tested within the same condition simultaneously, that will definitely emphasize their safety profile.
Author Response
Response to Reviewer 1 Comments
Dear Reviewer:
We sincerely thank you for your comments. They are helpful for revising and improving our manuscript. We have studied all the comments carefully and made corrections which we hope meet with approval. All the revisions were highlighted in the uploaded file. Our responses one by one to your comments are listed as follows (the replies are highlighted in red):
Comment #1: Line 55-56. “Studies indicate that this strain in SSF can effectively enhance steroid yield and abundance.” Which study? Please provide references.
Reply: Yes, we have added reference citations (Line 65-67).
Comment #2: Line 56-57. “The excellent production potential of the strain SYfxl23.3 has attracted widespread attention from researchers.”
Reply: Yes, this sentence was indeed misstated, and we have made a deletion and formulation change.
评论 #3: 根据作者的说法,这种菌株可能是一种新菌株 [第 78-80 行];这种特定菌株如何吸引其他研究人员?请澄清。
回答: 是的,根据我们的文献研究,镰刀菌 (Fusarium spinosum spp.) 在固态发酵中可有效提高类固醇的产量和丰度。因此,Fusarium spinosum spp. 的潜在应用受到了广泛关注(第 67-68 行)。
评论 #4: 第 192-194 行。“计算了在不同作用时间抑制化合物 1-4 对 Hala、A549 和 Mad-mb468 细胞的 50% (IC50) 值的药物浓度,如图 3d 和表 3 所示。”- 请改写它,因为它很难理解。
回复:是的,我们已经修改了它(第 205-207 行)。
评论 #5:第 305 行。菌株名称错误;请更正
回复:是的,我们已经更正了(第 316 行)。
评论 #6:第 307-309 行。“此外,它还含有多种经济效益的产品,如 β-谷甾醇、菜油甾醇和豆甾醇。”请证明你的陈述是合理的。
回答: 是的,因为在我们的预制成分分析中,我们发现发酵产物含有 β-谷甾醇、菜油甾醇和豆甾醇,然后我们在这里做出了这个声明。但是,这些内容被用作另一篇文章的提交,因此我们进行了删除。
评论 #7: 优化发酵条件。对绿色化学原理的关注,例如在固态发酵中使用尖孢镰刀菌菌株 SYfxl23.3,标志着制药的进步方法。然而,虽然该菌株的潜力得到了证明,但关于优化和扩大这种发酵过程的讨论有限。请发表评论。
回复:是的,我们在第 320-329 行添加了相关讨论,如下所示:
一些研究人员现在已经筛选了另一种镰刀菌属,通过优化培养基和其他添加剂,新培养基比传统培养基具有更高的薯蓣皂苷元生产率。内生刺镰刀菌 C39 被证明可有效将日本山药的皂苷转化为薯蓣皂苷元,在固态发酵 15 天后,薯蓣皂苷元浓度增加了 62.67%,此外还鉴定了 32 种化合物。同样,我们研究的棘镰刀菌菌株具有很大的潜力,可以通过进一步优化发酵条件和扩大发酵规模来提高生产效率,为甾体生物活性资源提供有力支持。也为未来的天然产物生产和药物开发提供了重要的基础和方向。
评论 #8: 安全概况。对于安全性,在细胞毒性和抗增殖测定实验期间,我想知道作者是否针对任何正常的健康细胞系测试了分离的化合物。如果它们同时在相同条件下进行测试,那肯定会强调它们的安全性。
回复: 是的,我们测试了 1-脱氢薯蓣酮对人胚胎肾细胞 (HEK9 细胞) 的毒性。结果表明,当以 400 μM 的浓度施用时,1-脱氢薯蓣酮对 HEK9 细胞没有显着毒性(图 S-6),我们也将其添加到手稿和补充材料中(第 211-214 行)。
图 S-6 1-脱氢薯蓣酮在体外对 HEK-293(人胚胎肾细胞)的细胞毒活性。
我们再次感谢您在手稿上花费的时间和精力。您的宝贵建议和评论对我们的修订有很大帮助。

Reviewer 2 Report
Comments and Suggestions for Authors
The authors of the manuscript “Potent Anticancer Activity of 1-Dehydrodiosgenone from the Product of Microbial Transformation of Steroid Saponins” describe the isolation of an endophytic fungus from Dioscorea, identified as Fusarium oxysporum through both morphological and molecular characteristics. The authors successfully isolated four steroidal compounds, such as diosgenin, smilagenone, yamogenin, and 1-dehydrodiosgenone, from Fusarium oxysporum. They subsequently screened these compounds for their antitumor activity against Hala, A549, and Mad-mb468 cells, and confirmed the antitumor properties of 1-Dehydrodiosgenone. Experiments demonstrated increased cell death through the intrinsic apoptotic pathway, with elevated levels of apoptosis-related proteins Bax and Bad, and activation of the caspase cascade. Safety tests on mice showed no significant inflammatory responses or toxic effects on major organs.
Specific comments:
Abstract:
L17: Please indicate the IC50 value of 1-Dehydrodiosgenone.
Keywords: Please avoid keywords from the title.
Introduction: Please update the information with recent articles. Please provide information on the microbial production of steroids.
L25-27: Please rewrite the sentence.
L29-30: Please update the information and cite a recent article [8].
L37-38: Citation required.
L43. Please update the information and cite a recent article [12].
L46-47: ‘various improvement methods’, please indicate the advancements and challenges.
L55: ‘Dioscorea’ Please italicize the species name throughout the text, including references.
L59: Please expand the abbreviation at first use ‘SSF’. The abstract is distinct from the introduction.
Results: Please indicate the yield of diosgenin, smilagenone, yamogenin, and 1-dehydrodiosgenone.
L196-199: Please move it to discussion.
L206: Figure 3d. Mean separation is required.
L259: Please improve the quality of Figure 4a.
Discussion: Please discuss the results of the steroids produced from Fusarium species, as well as the content of diosgenin, smilagenone, yamogenin, and 1-dehydrodiosgenone.
L305: Aspergillus terreus? Should be Fusarium oxysporum
Materials and Methods:
L382: spp., (non-italic).
L383: Please include the procedure for the isolation of SYfxl23.3 and its morphological characterization.
Conclusion:
L546: Fusarium spp.? Should be Fusarium oxysporum
Author Response
Response to Reviewer 2 Comments
Dear Reviewer:
We sincerely thank you for your comments. They are helpful for revising and improving our manuscript. We have studied all the comments carefully and made corrections which we hope meet with approval. All the revisions were highlighted in the uploaded file. Our responses one by one to your comments are listed as follows (the replies are highlighted in red):
Comment #1: Please indicate the IC50 value of 1-Dehydrodiosgenone.
Reply: Yes, we have added the IC50 value for 1-Dehydrodiosgenone in the abstract (Line 16-18).
Comment #2: Keywords: Please avoid keywords from the title.
Reply: Yes. We have corrected that (Line 23).
Comment #3: Introduction: Please update the information with recent articles. Please provide information on the microbial production of steroids.
Reply: Thank you for pointing this out. We have added relevant statements to the introduction and updated citations to articles from recent years (Line 59-65).
Comment #4: L25-27: Please rewrite the sentence.
Reply: Yes, we have corrected that (Line 27-28).
Comment #5: Please update the information and cite a recent article [8].
Reply: Yes, we have corrected that (Line 30-31).
Comment #6: L37-38: Citation required.
Reply: Yes, we have added a literature citation here (Line 38-39).
Comment #7: L43. Please update the information and cite a recent article [12].
Reply: Yes, we have corrected it (Line 45).
Comment #8: L46-47: ‘various improvement methods’, please indicate the advancements and challenges.
Reply: Yes, we have made the change of the revised manuscript. The changes are the following (Line 45-51).
To address this problem, researchers have proposed a variety of improvement methods, including acid recycling, ionic liquids to replace the use of conventional acids, solid acid process catalysis and photocatalysis. Although these improvements have reduced pollution to some extent, there are still some technical bottlenecks. For ex-ample, the preparation of ionic liquids and solid acids is complicated, and the thermal hydrolysis reaction must be carried out at high temperature and pressure, resulting in high energy consumption. Photocatalysis suffers from slower reaction rates, lower catalyst efficiency and greater dependence on light sources. Moreover, issues such as high costs and low efficiency have restricted the large-scale industrial application of these methods.
Comment #9: L55: ‘‘Dioscorea’ Please italicize the species name throughout the text, including references.
Reply: Yes, we have corrected it.
Comment #10: L59: Please expand the abbreviation at first use ‘SSF’. The abstract is distinct from the introduction.
Reply: Yes, we added this to the introduction where SSF first appeared (Line 66).
Comment #11: Results: Please indicate the yield of diosgenin, smilagenone, yamogenin, and 1-dehydrodiosgenone.
Reply: Yes, we have added this to the revised manuscript.
Comment #12: L196-199: Please move it to discussion.
Reply: Yes, we have corrected it (Line 345-349).
Comment #13: L206: Figure 3d. Mean separation is required.
Reply: Yes, we have corrected it.
Comment #14: Discussion: Please discuss the results of the steroids produced from Fusarium species, as well as the content of diosgenin, smilagenone, yamogenin, and 1-dehydrodiosgenone.
Reply: Yes, we have made changes to the discussion section in the revised manuscript (Line 435-438).
Comment #15: L305: Aspergillus terreus? Should be Fusarium oxysporum
Reply: Yes, we have corrected it (Line 316).
Comment #16: L382: spp., (non-italic).
Reply: Yes, we have corrected it.
Comment #17: L383: Please include the procedure for the isolation of SYfxl23.3 and its morphological characterization.
Reply: Yes, we have corrected it. The procedure for isolating SYfxl23.3 has been updated in the revised manuscript and its morphological characterization is presented in the Results section (Line 404-414). As follows:
The surface of the Dioscorea rhizomes was first washed three times with sterile water. The Dioscorea rhizomes were immersed in 75% alcohol for 3 minutes. The Dioscorea rhizome was then removed in an ultra-clean bench and the Dioscorea rhizome was crushed with a mortar and pestle after the alcohol had evaporated. Finally, 10×, 102×, 103×, 104×, 105×, 106×, 107× and 108× were diluted with sterile water and inoculated (100µL) onto potato dextrose agar (PDA) using the coated plate technique. Each gradient was repeated three times and incubated in a constant temperature incubator (28°C) for 3-4 d. Purification was repeated until single colonies were obtained.
Comment #18: L546: Fusarium spp.? Should be Fusarium oxysporum
Reply: Yes, we have corrected it (Line 577).
We thank you again for your time and effort on the manuscript. Your valuable suggestions and comments have been of great help in our revisions.

Reviewer 3 Report
Comments and Suggestions for Authors
The manuscript “Potent Anticancer Activity of 1-Dehydrodiosgenone from the Product of Microbial Transformation of Steroid Saponins” by Li et al. addresses the issue of obtaining steroids using microbial transformation. The article is very well planned and presented. It addresses a topic that may have great therapeutic potential.
The introduction sufficiently introduces the issues of the article. However, I believe that lines 60-62 should not be placed in the place where the aim of the work is described. The indicated fragments already present the results of the article.
The methodology is sufficiently described, which allows us to state that the research was performed correctly. I have no comments on this part.
The results and discussion are presented correctly. The results are described in detail, and the discussion explains the reasons for obtaining such results.
The conclusions correctly summarize the obtained results. It would also be possible to mention here the authors' future plans for the isolated steroids.
The bibliography requires minor additions.
Author Response
Response to Reviewer 3 Comments
Dear Reviewer:
We sincerely thank you for your comments. They are helpful for revising and improving our manuscript. We have studied all the comments carefully and made corrections which we hope meet with approval. All the revisions were highlighted in the uploaded file. Our responses one by one to your comments are listed as follows (the replies are highlighted in red):
Comment #1: I believe that lines 60-62 should not be placed in the place where the aim of the work is described.
Reply: Yes, we have made changes in the new manuscript. The modifications are as follows:
In this study, we successfully isolated an endophytic strain Fusarium oxysporum strain from the Dioscorea rhizome and named it SYfxl23.3. Systemic steroidal precursors with broad-spectrum antitumor activity and low toxicity were produced by using SYfxl23.3 and SSF. We screened and identified four steroidal compounds and analyzed their chemical properties. The anti-tumor activity was screened using cellular assays and focused on the potential application of 1-Dehydrodiosgenone. In addition, the safety of the compound to ensure that its therapeutic potential would not result in toxic side effects. This study provides a theoretical basis for the application of 1-Dehydrodiosgenone as an antitumor drug precursor and to promote it as a potential antitumor drug candidate (Line 69-77).
Comment #2: The conclusions correctly summarize the obtained results. It would also be possible to mention here the authors' future plans for the isolated steroids.
Reply: Yes, we have added this to the revised manuscript (Line 589-591).
Comment #3: The bibliography requires minor additions.
Reply: Yes, we have supplemented the references.
We thank you again for your time and effort on the manuscript. Your valuable suggestions and comments have been of great help in our revisions.
